# Functional Recellularization of Acellular Rat Liver Scaffold by Induced Pluripotent Stem Cells: Molecular Evidence for Wnt/B-Catenin Upregulation

**DOI:** 10.3390/cells10112819

**Published:** 2021-10-20

**Authors:** Nesrine Ebrahim, Omnia A. M. Badr, Mohamed M. Yousef, Amira Hassouna, Dina Sabry, Ayman Samir Farid, Ola Mostafa, Hajir A. Al Saihati, Yasmin Seleem, Eman Abd El Aziz, Ahmed Hassan Khalil, Ahmed Nawar, Ahmed A. Shoulah, Mohammad Aljasir, Amira Zaki Mohamed, Mohamed El-Sherbiny, Nehal M. Elsherbiny, Mohamed Ahmed Eladl, Nicholas Robert Forsyth, Rabab F. Salim

**Affiliations:** 1Department of Histology and Cell Biology, Faculty of Medicine, Benha University, Banha 13511, Egypt; nesrien.salem@fmed.bu.edu.eg (N.E.); mohamed_jo63@yahoo.com (M.M.Y.); ola.mostafa.moez@gmail.com (O.M.); 2Stem Cell Unit, Faculty of Medicine, Benha University, Banha 13511, Egypt; 3Department of Genetics and Genetic Engineering, Faculty of Agriculture, Benha University, Banha 13511, Egypt; omnia.badr@fagr.bu.edu.eg; 4School of Public Health and Interdisciplinary Studies, Faculty of Health and Environmental Sciences, AUT University, Auckland 1010, New Zealand; amira.hassouna@aut.ac.nz; 5Department of Medical Biochemistry and Molecular Biology, Faculty of Medicine, Cairo University, Cairo 12613, Egypt; dinasabry@kasralainy.edu.eg; 6Department of Medical Biochemistry and Molecular Biology, Faculty of Medicine, Bader University in Cairo, Cairo 11562, Egypt; 7Department of Clinical Pathology, Faculty of Veterinary Medicine, Benha University, Banha 13511, Egypt; ayman.samir@fvtm.bu.edu.eg; 8Department of Clinical Laboratory Sciences, College of Applied Medical Sciences, University of Hafr Albatin, Hafar Al Batin 39524, Saudi Arabia; hajirsh@uhb.edu.sa; 9Department of Clinical Pharmacology, Faculty of Medicine, Benha University, Banha 13511, Egypt; yasmeen.seleem@fmed.bu.edu.eg (Y.S.); drimanpharma@gmail.com (E.A.E.A.); 10Department of Surgery & Radiology, Faculty of Veterinary Medicine, Benha University, Banha 13511, Egypt; ahmed.khalil@fvtm.bu.edu.eg; 11Department of General Surgery, Faculty of Medicine, Benha University, Banha 13511, Egypt; Nowar79@yahoo.com (A.N.); ahmedsho82@yahoo.com (A.A.S.); 12Department of Medical Laboratories, College of Applied Medical Sciences, Qassim University, Buraydah 52571, Saudi Arabia; mjasr@qu.edu.sa; 13Department of Microbiology, Faculty of Science, Tanta University, Tanta 31527, Egypt; mira26@hotmail.com; 14Department of Basic Medical Sciences, College of Medicine, AlMaarefa University, Riyadh 71666, Saudi Arabia; msharbini@mcst.edu.sa; 15Department of Anatomy, Mansoura Faculty of Medicine, Mansoura University, Mansoura 35516, Egypt; 16Department of Biochemistry, Faculty of Pharmacy, Mansoura University, Mansoura 35516, Egypt; 17Department of Pharmaceutical Chemistry, Faculty of Pharmacy, University of Tabuk, Tabuk 47512, Saudi Arabia; 18Department of Basic Medical Sciences, College of Medicine, University of Sharjah, Sharjah 27272, United Arab Emirates; 19Guy Hilton Research Laboratories, School of Pharmacy and Bioengineering, Faculty of Medicine and Health Sciences, Keele University, Newcastle ST5 5BG, UK; n.r.forsyth@keele.ac.uk; 20Department of Medical Biochemistry and Molecular Biology, Faculty of Medicine, Benha University, Banha 13511, Egypt

**Keywords:** decellularization, recellularization, Wnt/*β*-catenin pathway, liver regeneration, iPSC, regenerative medicine

## Abstract

Background. Liver transplantation remains the only viable therapy for liver failure but has a severely restricted utility. Here, we aimed to decellularize rat livers to form acellular 3D bio-scaffolds suitable for seeding with induced pluripotent cells (iPSCs) as a tool to investigate the role of Wnt/β-catenin signaling in liver development and generation. Methods. Dissected rat livers were randomly divided into three groups: I (control); II (decellularized scaffolds) and III (recellularized scaffolds). Liver decellularization was established via an adapted perfusion procedure and assessed through the measurement of extracellular matrix (ECM) proteins and DNA content. Liver recellularization was assessed through histological examination and measurement of transcript levels of Wnt/β-catenin pathway, hepatogenesis, liver-specific microRNAs and growth factors essential for liver development. Adult rat liver decellularization was confirmed by the maintenance of ECM proteins and persistence of growth factors essential for liver regeneration. Results. iPSCs seeded rat decellularized livers displayed upregulated transcript expression of Wnt/β-catenin pathway-related, growth factors, and liver specification genes. Further, recellularized livers displayed restored liver-specific functions including albumin secretion and urea synthesis. Conclusion. This establishes proof-of-principle for the generation of three-dimensional liver organ scaffolds as grafts and functional re-establishment.

## 1. Introduction

Liver diseases are considered one of the most serious global medical problems accompanied by a high rate of morbidity and mortality [1]. Essential for survival, featuring a range of key functions [2], numerous etiologies have emerged that can trigger liver dysfunction leading to chronic liver disease or acute liver failure. The treatment of choice for end-stage liver disease, liver metabolic disease, and hepatocellular carcinoma is liver transplantation [3]. The increased success rate of liver transplantation is met by the challenge of reducing the number of accessible donors. Moreover, when considering the high cost of liver transplantation, major complications [4], the long-term requirement for immunosuppressive drugs, exploiting alternative therapies for end-stage liver diseases are critically needed [5,6].

Cell-based treatments and liver tissue engineering have emerged as promising alternatives for the treatment of liver disease in the last decade. Tissue-engineered livers are described by the use of a biomaterial scaffold to provide the complex extracellular matrix (ECM) of the liver and to support the subsequent seeding and survival of biologically competent cells [7]. The ECM has a central role in different cellular signaling pathways within the liver including differentiation [8], growth and proliferation [9,10], and angiogenesis [11]. The liver-specific ECM is therefore a key component of any three-dimension (3D) structure proposed for long-term hepatocyte survival and success of tissue-engineered liver grafts. In principle, liver ECM scaffolds can be obtained from allogeneic or xenogeneic donor livers, decellularized to remove cellular immunogenic elements (DNA, intracellular protein), while preserving the composition of ECM and the ultrastructure [12]. The combination of a decellularized 3D liver ECM scaffold with a suitable cell line could in principle generate a bioengineered liver [13].

Induced pluripotent stem cells (iPSCs) self-renew have a pluripotent differentiation potential, and create the opportunity to manufacture a wide range of patient-specific cells [14]. iPSCs have been confirmed to have endodermal differentiation potential allowing them to differentiate into hepatocyte-like cells with culturing in suitable culture conditions [15]. Most of the differentiation techniques comprise numerous extra cytokines and growth factors, like hepatocyte growth factor (HGF), epidermal growth factor (EGF), fibroblast growth factor (FGF), and oncostatin M (OSM). iPSCs can differentiate into hepatocytes phenotype through three stages (initiation, differentiation, and maturation) [16]. The control of these differentiation routes lies under Nodal, Wnt/ b-catenin, and TGF-ß signaling pathways, which work in synergy to specify definitive endoderm, hepatic specification, and maturation. In this context, the initiation phase is characterized by iPSCs differentiation into endodermal germ cells through nodal signaling pathway where high doses induce endoderm [17]. Nodal signaling induces expression of endodermal transcription factors; e.g., Sox17 and s Foxa1–3 [18], which consecutively regulate a transcriptional cascade driving commitment to the endodermal lineage. The second stage, differentiation, sees hepatoblast specification through Wnt/b-catenin reflecting the Wnt/b-catenin pathway autonomy in the endodermal cell linage to induce hepatic specification, proliferation, and differentiation. Continued Wnt /b- catenin regulates the bi-potential differentiation of hepatoblasts into either hepatocytes or biliary epithelial cells. Besides its transcriptional functions, the β-catenin can also contribute to cell-cell adhesion through its association with E-cadherin at the plasma membrane [19]. Thus, it was proofed that the β-catenin has two pools, one for the transcriptional activation of Wnt target genes, and the other is cadherin-based for adherent’s cell junctions. Thus, β-catenin’s, via its role in cell-cell adhesion, exert a significant function in hepatocyte maturation (third stage) [20]. Canonical Wnt signaling was inactive in the absence of Wnt ligands (Wnt off). In this state, β-catenin is located in adherent junctions and the cytoplasm, where it becomes phosphorylated by the destruction complex (comprising adenomatous polyposis coli protein (APC), Axin, casein kinase I isoform-α (CK1α), and glycogen synthase kinase 3β (GSK3β) and is targeted for proteasomal degradation. Wnt signaling was active in the presence of Wnt ligands (Wnt on), which bind to the FZD–LRP5–LRP6 co-receptor. Subsequent LRP6 phosphorylation results in Axin and Dishevelled (DVL) recruitment, which blocks Axin-mediated phosphorylation of β-catenin and thereby prevents β-catenin degradation, enabling its accumulation and nuclear translocation. In the nucleus, β-catenin binds to diverse co-effectors, regulating the expression of genes involved in different cellular processes [20].

Collectively, the Wnt/β-catenin signaling is considered amongst the major signal transduction pathways in liver cell biology. This pathway directs the key physiological actions inherent to the liver including development and regeneration processes via transduction across proliferation, differentiation, zonation, adhesion, metabolism, and apoptosis. Developmentally, Wnt/β-catenin signaling regulates multiple stages including competence, hepatic induction, expansion, and morphogenesis [21].

The objectives of the present study were to establish a functional 3D liver bioscaffold following decellularization and iPSC-based recellularization protocols and using gene analysis, to identify wnt/β-catenin signaling pathway activation mechanisms in iPSC hepatic stepwise differentiation.

## 2. Materials and Methods

### 2.1. Experimental Animals

Male albino rats (180–200 g), 6 weeks old, were obtained from Benha University, Faculty of Veterinary Medicine. All of the animals were kept in clean cages and provided with a normal diet and clean water ad libitum. Light (12 h light-dark cycle starting at 8:00 a.m.) and room temperature (23 ± 3 °C) were also adjusted in the animal’s habitat. This work was conducted in strict conformity with the National Institutes of Health’s Guide for the Care and Use of Laboratory Animals (NIH publication No. 85–23, revised 2011). The Faculty of Medicine, Benha University, Egypt’s institutional review board for animal experimentation authorized all protocols (BUFM 3 January 2018).

### 2.2. Experimental Design and Treatment Protocol

Thirty-five rats were randomly divided into 3 groups:**Group I (control group; *n =* 7):** Following sacrifice, livers were dissected and processed for analysis without any intervention.**Group II (decellularized group; *n =* 7):** Following sacrifice, livers were surgically removed with preservation of portal vein, inferior vena cava, and bile duct. Livers were immediately decellularized by perfusion.**Group III (recellularized groups; *n =* 21)**: Following sacrifice, livers were surgically removed, as above, and then decellularised by perfusion. Following decellularization, liver bioscaffolds were seeded with iPSCs (6 × 10^6^ at 4 mL/min flow) then equally divided into three groups with different culture periods:
(a)**Subgroup IIIa:** 4 days of the recellularization process.(b)**Subgroup IIIb:** 14 days of the recellularization process.(c)**Subgroup IIIc:** 24 days of the recellularization process.

### 2.3. Surgical Procedure for Liver Harvesting

A longitudinal abdominal incision was used to expose the lower abdominal cavity, the liver, and the rib cage. The superior vena cava was bisected as adjacent to the atrium as possible, along with cardiac and falciform ligaments. The diaphragm was cautiously separated from the esophagus to isolate it from the diaphragm and liver. The adipose tissue layers surrounding the portal vein were separated cautiously to expose the vein and its branches. The lateral branches were then ligated with silk suture 6–0 and cut near to the intestines (distal to the liver). The common bile duct was then dissected and transected at the duodenum. The portal vein was transected 1.5–2 cm away from the liver. The infra-hepatic vena cava was then placed below the right lobe of the liver, cautiously dissected, and bisected without any damage to the liver lobe. Before removing the whole liver, careful confirmation was made that no additional attachments to the liver were present, before gently removing the liver holding it by the diaphragm. The portal vein was then cannulated with a 20 G cannula before being infused with heparinized 0.9% saline. Following saline infusion, the liver was removed, weighed, and then either preserved at 4 °C or frozen at −80 °C for a minimum of 12 h prior to decellularization [13].

### 2.4. Bioreactor Perfusion System

This system consisted of a vessel (Glass Ball Spinner, 250 mL, Bellco Biotechnology Inc., Vineland, NJ, USA), a peristaltic pump (Master flex, L/S with Master Flex L/S easy load pump head, Cole Parmer, Vernon Hills, IL, USA), Pulse dampener (Cole Parmer), oxygenator, and syringe infusion pump. The system was positioned in an incubator for control of temperature, and the oxygenator connected to the atmospheric gas mixture. The graft was continuously perfused through the portal vein at 4 mL/min with continuous oxygenation that delivered an inflow partial oxygen tension of ~300 mmHg.

### 2.5. Whole-Organ Liver Decellularization

Frozen livers were thawed at 4 °C and then perfused with PBS overnight to eliminate blood through the portal vein at 4 mL/min. Decellularization was attained by perfusion of the native liver by sodium dodecyl sulfate (SDS; Sigma, Aldrich, Egypt) in deionized water for a total of 72–96 h starting with 0.01% SDS for 24 h, followed by 0.1% SDS for another 24 h, which was followed by 1% SDS for 48 h or more. The liver was then washed with deionized water for 15 min and then 1% Triton X-100 (Sigma) for 30 min. Decellularized livers were then washed with PBS for 1 h before being sterilized in 0.5% peracetic acid (Sigma) in PBS for 3 h. The liver bio-scaffold was washed via perfusion extensively with sterile PBS and preserved in PBS supplemented with antibiotics at 4 °C for up to 7 days [22].

### 2.6. Measurement of DNA Content

Using Genomic DNA Mini Kit (Invitrogen), genomic DNA was extracted from 15 mg of tissue (dry weight). DNA concentration was measured at 260 nm and normalized to the initial dry weight of the samples utilizing a NanoDrop spectrophotometer (ND-2000c; Thermo, USA) [23].

### 2.7. Glycosaminoglycan (GAGs) Assay

Blyscan GAG kit (B1000; Biocolor, Carrickfergus, UK) was used to measure sulfated GAGs. Samples were lyophilized, weighed, and incubated for 3 h at 65 °C with papain (150 g/mL). After adding the Blyscan dye reagent, the samples were homogenized for 30 min before being centrifuged for 10 min (10,000× *g*). A dissociation reagent was used to dissolve the pellets, and the absorbance was measured at 650 nm.

#### 2.7.1. iPS Cells (iPSCs) Generation

iPSCs were generated according to Okita et al., (2011) The iPSCs line used in this study was generated from dermal fibroblasts derived from a skin biopsy taken from a healthy rat [24]. 2 cm^3^ of rat skin were collected and then sliced into minor pieces via sterile scalpels, and then incubated at 37 °C for 12 h in collagenase type II (3 mg/mL) (Sigma Aldrich, Merck KGaA, Darmstadt, Germany). The samples then were strained into a 15 mL falcon tube to remove tissue debris and washed with phosphate buffer saline (PBS). Cells were pelleted by centrifugation at 161× *g* for 4 min. The pelleted cells were re-suspended in RPMI 1640 medium containing 10% FBS and plated in a T12.5 mL flask at 37 °C in 5% CO_2_ for 3 days without manipulation to allow fibroblasts to adhere. Culture medium was changed every 2 days until confluence was reached after 7−10 days. The fibroblasts were re-programmed into iPS cells using an Amaxa 4D-Nucleofector (P2 Primary Cell Kit V4XP-2012, Program FF-135; Lonza, Basel, Switzerland) with proplasmid^TM^ from GenScript ProBio containing OCT4, SOX2, c-Myc and LIN28. One day prior to transfection, 5 × 10^5^ cultured cells were plated in 1 mL complete growth medium and allowed to grow until the cells were 50–70% confluent at the time of transfection. The four essential transfection factors, Oct3/4, SOX2 and c-Myc and LIN28, were transfected in a single plasmid using the Nucleofactor kit (Lonza) according to the manufacturer’s instructions. Six days after transfection, cells were replated on Matrigel coated plates (BD Biosciences, San Jose, CA, USA) and fed with the serum-free human iPSC media consisting of Pluriton medium (Stemgent) supplemented with 20% (vol/vol) mTeSR-1 maintenance media (Stemcell Technologies, Vancouver, BC, Canada). One to three weeks after transfection, the cells began to form iPSC-like colonies, and at 3 to 6 weeks, colonies were picked on the basis of size and morphology. For iPSCs characterization, RT-PCR was performed for pluripotent genes (OCT4, SOX2, c-Myc, TERT & NANOG) in different 5 passages (passage 3, 6, 8, 10, and 13).

#### 2.7.2. In Vitro Pluripotency Assay—Embryoid Body Formation and Spontaneous Differentiation

Embryoid bodies were generated from iPSCs by passaging the colonies as small clusters and culturing them in DMEM-F12 medium supplemented with N2 and B27 in the absence of bFGF in low adherent six well petri dishes. The EBs were allowed to adhere on matrigel after 7 days and assessed for trilineage markers by RT-PCR. Differentiated cells were analyzed for markers of three germ layers. For mesodermal differentiation we detect (CD31, RUNEX1 & ACTA2) and for ectodermal differentiation we detect (GATA4 & GATA6). While, for endodermal differentiation we used (TUBB3 & PAX6).

### 2.8. Perfusion Recellularized Liver Bioreactor

The acellular liver bio-scaffold was sterilized via 0.5% peracetic acid (PPA) sterilization as the PAA method considered a potent method for sterilization of biological scaffolds then the acellular bio-scaffold were transferred to the perfusion system. The perfusion system was positioned in an incubator for control of temperature, and the oxygenator was linked to an atmospheric gas mixture. The acellular bio-scaffold was linked to the perfusion system by a portal vein cannula. The inferior vena cava was left open for outlet and the superior vena cava was ligated. The acellular bio-scaffold was retained in continuous perfusion via the portal vein. Cuffs placed in the portal vein and inferior vena cava permitted easy manipulation of the acellular rat liver. The acellular bio-scaffold was continuously perfused through the portal vein at 4 mL/min with continuous oxygenation that delivered an inflow partial oxygen tension of ~300 mmHg. Fluid media underwent a daily cycle of complete perfusion. Cells were seeded by intra-portal multistep infusion according to previous reports [14]. Cell delivery was achieved using a controlled syringe infusion pump connected directly to the perfusion tubing. After 30–60 min perfusion with medium, a total of 6 × 10^6^ iPSC were infused in four steps at 15 min intervals, each step saw 1.5 × 10^6^ cells suspended in 0.5 mL of medium perfused over 5 min. Viability was assessed by the trypan blue exclusion test and was routine >90%. To estimate the engraftment efficacy, the perfusate was collected, and the viability and number of cells which not retained in the bio-scaffold was estimated with a trypan blue exclusion and hemacytometer. The total number of cells retained in the bio-scaffold denoted the difference between the initial number of cells seeded and the number of cells present in the perfusate after seeding.

#### 2.8.1. Differentiation of iPSCs into Hepatocytes within the 3D Acellular Liver Bio-Scaffold

To induce differentiation of iPSCs toward a hepatocyte phenotype within acellular bio-scaffolds, we adopted a modified version of a previously published protocol [14].

**Stage 1 (endodermal induction).** The iPSCs were initially perfused within a media consisting of RPMI (Invitrogen, Carlsbad, CA, USA), 0.5% Penicillin/Streptomycin (Millipore, Billerica, MA, USA), 1X B-27 w/o insulin supplement (Invitrogen, Carlsbad, CA, USA), 0.5% Non-Essential Amino Acids (Millipore, Billerica, MA, USA), 10 ng/mL BMP4 (R&D Systems, Minneapolis, MN, USA), 100 ng/mL Activin A (R&D Systems, Minneapolis, MN, USA), and 20 ng/mL FGF2 (BD, Franklin Lakes, NJ, USA) for 2 days. After 48 h the perfused media was switched to one without FGF2 and BMP4 for a further 2 days; at this stage, the population is considered as iPS-Heps cells. (**Stage 2, definitive endoderm).****Stage 3 (hepatic specification):** The iPS Heps cells were then perfused for a further 10 days within a defined medium containing 45% DMEM low glucose 1 g/L (ThermoFisher Scientific, Waltham, MA, USA), 10% CTS KnockOut SR XenoFree Medium (ThermoFisher Scientific, Waltham, MA, USA), 45% F-12 (ThermoFisher Scientific, Waltham, MA, USA), 0.5% Non-Essential Amino Acids (ThermoFisher Scientific, Waltham, MA, USA), 50 ng/mL HGF (Kindly provided by George Michalopoulos) 0.5% L-glutamine (ThermoFisher Scientific, Waltham, MA, USA), and 1% DMSO (Sigma-Aldrich, Saint Louis, MO, USA). The medium was changed every 48 h.**Stage 4 (hepatocytes maturation):** Following the specification, the cultured iPS-Heps were maintained under perfusion within acellular bio-scaffolds for a further 10 days. Maturation media consisted of specification media further supplemented with 0.1% of Gentamicin/Amphotericin-B (ThermoFisher Scientific, Waltham, MA, USA), 1% of Penicillin/Streptomycin (ThermoFisher Scientific, Waltham, MA, USA), 0.1% of Ascorbic Acid (Sigma-Aldrich, Saint Louis, MO, USA), 0.5 µM Dexamethasone (Sigma-Aldrich, Saint Louis, MO, USA), 0.1% of Bovine Serum Albumin Free of Fatty Acids, 0.1% of Transferrin, 0.1% of Hydrocortisone, 0.1% of Insulin (HCM Bullet Kit, ThermoFisher Scientific, Waltham, MA, USA), 20 µM of Palmitic Acid (Sigma-Aldrich, Saint Louis, MO, USA), 100 µM of Urso deoxycolic acid (Sigma-Aldrich, Saint Louis, MO, USA), 30 µM of Oleic Acid (Sigma-Aldrich, Saint Louis, MO, USA), 1× of Cholesterol and 20 µM of Rifampicin (Sigma-Aldrich, Saint Louis, Missouri) (ThermoFisher Scientific, Waltham, MA, USA).

#### 2.8.2. Detection of Albumin and Urea

Perfused culture media was collected daily for functional evaluation. Albumin and urea concentrations were determined by Rat Albumin ELISA kit (ab108790), and Urea Assay Kit (ab83362), respectively, following the manufacturer’s instructions. Absorbance was measured in a microplate reader (Epoch BioTek, Winooski, VT, USA). Concentrations were presented as µg/mL.

### 2.9. Quantitative Real-Time PCR (RT-PCR)

TRIzol (Invitrogen, Waltham, MA, USA) was used to extract total RNA according to the manufacturer’s instructions. A Nano-Drop 2000 C spectrophotometer was used to determine the concentration and the purity of the extracted RNA (Thermo Scientific, Waltham, MA, USA). All samples had an RNA purity of >1.9 at the A260/A280 absorbance ratio. On a 2% agarose gel, the integrity of the RNA was checked using gel electrophoresis image (Gel Doc. Bio-Rad, Hercules, CA, USA). SensiFast cDNA synthesis kits (Bioline, Meridian Bioscience, London, UK) were used to make complementary DNA (cDNA) according to the manufacturer’s instructions. NCode VILO miRNA cDNA Synthesis Kit (Invitrogen) was used for cDNA Synthesis from miRNAs following the manufacturer’s recommendations.

Quantitative PCR reaction was performed with Maxima SYBR Green/ROX qPCR (Thermo Scientific, USA). Primer pairs were designed using Primer Express 3 and verified with in silico PCR tool (http://insilico.ehu.eus/PCR/, accessed on 31 August 2021) as previously described by Abdelatty et al. [25]. Selected target and reference genes, were classified according to the stage of iPSCs differentiation into:**Initiation stage (endodermal specification):** FOXA1, Forkhead Box A1; FOXA2, Forkhead Box A2; SOX17, SRY-Box Transcription Factor 17.**Hepatoblasts specification:** Wnt ligands (Wnt 1, 2 3a, 7b, 10b), HHEX, Hematopoietically expressed homeobox; HLX, H2.0-like homeobox; Prox1, Prospero homeobox 1, FGF2, Fibroblast growth factor 2; BMP4, Bone Morphogenetic Protein 4; HGF, Hepatocyte Growth Factor; FGF4, Fibroblast growth factor 4; EGF, Epidermal growth factor; TGF-β, Transforming Growth Factor Beta 1; ALB, Albumin; AFP, Alpha-fetoprotein; CK19, Cytokeratin 19.**Hepatocytes specification & maturation**: C/EBPα, CCAAT/enhancer-binding protein alpha; HNF4a, Hepatocyte nuclear factor 4, alpha; PECAM1, platelet and endothelial cell adhesion molecule 1; EpCAM, Epithelial cell adhesion molecule; Beta-actin; miR122; miR148a; miR194; U6 RNA) were purchased from Genwez (South Plainfield, NJ, USA) (Table 1).

PCR reactions contained 500 ng cDNA, 12.5 μL Maxima SYBR Green Master Mix (Maxima SYBR Green qPCR, ThermoFisher Scientific, Waltham, MA, USA), 0·3 μmol L^−1^ of forward and reverse primer, 10 nmol L^−1^/100 Nm ROX Solution, the final volume was adjusted to 25 μL by adding nuclease-free water. The reaction was carried out in AriaMx Real-Time PCR (Agilent Technologies, Santa Clara, CA, USA) using a two-step protocol that included initial denaturation at 95 °C for 10 min, followed by 40 cycles of denaturation at 95 °C for 15 s and by annealing/extension at 60 °C for 60 s. At the end of the PCR, a melting curve technique was used, which consisted of heating at 95 °C for 30 s then 65 °C for 30 s, and 95 °C for the 30 s. The housekeeping gene *B-Actin* was used to standardize the expression levels of target genes while the expression of miRNAs was normalized relative to U6 RNA as an internal housekeeping reference gene. Relative gene expression ratios (RQ) between treated and control groups were measured by the formula: RQ = 2^−ΔΔCt^ [26].

### 2.10. Protein Analysis of ECM Competent and Growth Factors of Decellularized Bio-Scaffolds besides Protein Analysis of β-Catenin after Recellularization of 3D Bio-Scaffolds

Western blot analysis was performed as previously described [27,28]. Total cellular protein was extracted using RIPA Lysis Buffer (P0013B, Beyotime) and PMSF (ST506, Beyotime). The protein-transferred polyvinylidene difluoride (PVDF) membrane was probed with mouse monoclonal antibody against collagen type I (ab6308), rabbit polyclonal antibodies against laminin (ab7463) and hepatocyte growth factor (HGF) (ab216623), rabbit monoclonal antibodies against collagen type IV (ab236640), fibronectin (ab199056), basic fibroblast growth factor (bFGF) (ab171941), vascular endothelial growth factor (VEGF) (ab32152), insulin-like growth factor (IGF) (ab182408), total β-catenin (ab32572), activated β-catenin (ab246504), phosphorelated β-catenin (Y654,CTNNB1, A00004Y654), albumin (ab207327), Oct4 antibody(ab181557), alpha 1 Fetoprotein antibody (ab284388), SOX2 antibody(ab92494) & β-actin antibody(ab8226). All primary and secondary antibodies were purchased from Abcam (Cambridge, MA, USA).

Blots were incubated with secondary antibodies as follows: goat anti-mouse secondary antibody (ab205719) against collagen type I and collagen type IV, goat anti-rabbit secondary antibody (ab97051) against fibronectin, insulin-like growth factor (IGF) and zctivated β-catenin, and goat anti-rabbit secondary antibody (ab205718) against laminin, HGF, bFGF, VEGF, HRP-conjugated goat anti-rabbit secondary antibody (ab15007) against total β-catenin, goat anti-babbit IgG (H + L) secondary antibody, biotin conjugated (BA1003) against phosphorylated β-catenin, secondry goat anti-rabbit IgG H & L (HRP) (ab97051) against abumin, secondary goat anti-rabbit IgG H & L (HRP) (ab97051) against OCT4, goat anti-rabbit IgG H & L (HRP) (ab97051) against alfa fetoprotein, goat anti-rabbit IgG H & L (HRP) (ab205718) against SOX2, Nanog antibody (H-2): sc-374103 and mouse IgG Fc binding protein (m-IgG Fc BP): sc-525409 & goat anti-mouse IgG H & L (HRP) (ab205719) against Actin. Proteins were detected by ChemiDoc MP Imaging System (Bio-Rad, Hercules, CA, USA). Autoradiographs were quantified using Image-Pro Plus program version 6.0 (Media Cybernetics Inc., Bethesda, MD, USA)

### 2.11. Histological Study

Fresh, decellularized, and recellularized liver samples from days 4, 14, and 24 were excised and fixed in 10% buffered formol saline and processed as 4-6-µm-thick paraffin sections, followed by mounting on glass slides for H & E and Masson’s Trichrome staining. Histological sections were observed and analyzed microscopically (Leica DMR 3000; Leica Microsystem, Wetzlar, Germany) by two blinded experienced investigators [28].

### 2.12. Morphometric Study

The mean area (%) of collagen fibers identified with Masson’s trichrome were quantified in five non-overlapping fields from five images within each rat group. Quantification was performed using Image-Pro Plus program version 6.0 (Media Cybernetics Inc., Bethesda, MD, USA).

### 2.13. Scanning Electron Microscopy

Fresh, decellularized, and recellularized liver samples from days 4, 14, 24 were fixed in 2.5% glutaraldehyde in 0.1 M phosphate-buffered gluteraldehyde pH 7.4 at 4 °C for 2 h followed by three washes in PBS. The samples were post fixed in 1% osmic acid for 30 min, dehydrated with an ascending series of ethyl alcohol (30%, 50%, 70%, 90%, and absolute alcohol), and infiltrated with acetone for 30 min. Samples were then dried in a Critical Point Dryer (Tousimis Autosamdri-815 Coater), mounted on aluminum stubs, and coated with gold in an SPI-Module (Sputter Carbon/Gold Coater). Visualization was with a scanning electron microscope (JSM-6510 LV) (JEOL, Tokyo, Japan) at the electron microscope unit in the Faculty of Agriculture, EL-Mansoura University, El-Mansoura, Egypt. Pockets quantification was done using Image-Pro Plus program version 6.0 (Media Cybernetics Inc., Bethesda, MD, USA) as a minimum of 5 random figures on each sample.

### 2.14. Statistical Analysis

Statistical analysis was performed with SPSS for Windows (Version 20.0; SPSS Inc., Chicago, IL, USA). Differences between groups were evaluated using one-way analysis of variance (ANOVA; F) and the Kruskal Wallis test (χ^2^) to compare more than two groups of parametric and non-parametric data, respectively, followed by post-hoc analysis to detect differences in pairs. For each test, all data are expressed as the mean ± standard error mean (SEM), where a *p*-value < 0.05 was considered significant.

## 3. Results

### 3.1. Whole-Organ Rat Liver Decellularization and Characterization

First, we sought to confirm efficiency of the decellularization protocol in rat livers. The protocol was performed by exposing livers to freeze-thawing for a minimum of 12 h followed by perfusion of anionic detergent (SDS) and Triton X-100 as described above. The flow shear stress, coupled to detergent, allowed penetration into the hepatic sinusoid and the detachment of cells and debris. Indeed, macroscopic analysis of liver during decellularization displayed a progressive change in color to yellowish-brown, and ultimately transparency (Figure 1a–e). The perfused decellularization procedure generated an acellular scaffold that retained the gross morphology of the liver.

Additionally, DNA content was significantly reduced in decellularized scaffold (0.56 ± 0.067 µg/mg) compared to normal liver tissue (18.5 ± 2.22 µg/mg), demonstrating decreased nuclear material in the decellularized liver scaffold compared to the normal liver (*p* < 0.05). This was confirmed by results of gel electrophoresis of DNA content in the decellularized liver (DC) which revealed no visible bands compared to the control liver (C) (Figure 1f).

The efficacy of the decellularization process was also evaluated by estimation of GAG content. The quantity of remaining GAGs (71.4%) was relatively preserved with a significant change in the decellularized scaffold when compared to the normal liver (Figure 1g).

Regarding extracellular matrix (ECM), the acellular bio-scaffolds demonstrated retention of key ECM components, namely, collagen I (acellular bio-scaffold (0.26 ± 0.01 vs. native (0.33 ± 0.01), collagen IV (acellular bio-scaffold (0.26 ± 0.02) vs. native (0.27 ± 0.01), fibronectin (acellular bio-scaffold (0.22 ± 0.01) vs. native (0.26 ± 0.01) and laminin (native (0.33 ± 0.002) vs. acellular bio-scaffold (0.24 ± 0.02), and at levels that were consistent with native liver tissues (Figure 1h).

However, in contrast to matrix components, we observed stark significant reduction in levels of growth factors HGF, bFGF, VEGF, and IGF compared to native liver tissues where their average relative density were 0.15 ± 0.03, 0.19 ± 0.05, 0.15 ± 0.02, 0.09 ± 0.01, respectively compared to 0.79 ± 0.08, 0.99 ± 0.07, 0.80 ± 0.15, 0.71 ± 0.11, respectively in native liver tissues. In other words, only 19% of HGF, 20% of bFGF, 19% of VEGF, and 13% of IGF-1 were conserved in acellular bio-scaffold matrix when compared with normal liver (Figure 1i).

### 3.2. iPSCs Derivation and Characterization before Scaffold Implantation

Remarkably, 24 h after the reprogramming process, dermal fibroblasts exhibited marked morphological changes, from the initial spindle-shape to a more polygonal or epithelial-like cells morphology until confluence was reached after 7−10 days (Figure 2a,b). Three to six weeks after reprogramming colonies showing a typical iPSCs morphology appeared in culture (large nuclear to cytoplasmic ratio, and form domed colonies (Figure 2c). Based on colonies morphology, the calculated reprogramming efficiency was 0.95%. Fragments of ESC-like colonies after picking were able to form new colonies (clones). Furthermore, PCR analysis was performed to confirm the induction of endogenous pluripotency-associated markers including OCT4, SOX2, NANOG, TERT, and c-MYC. The markers of pluripotency remained unchanged from passages 3 to 13 (Figure 2d), suggesting the maintenance of an undifferentiated state of the iPSCs clones produced, thus confirming the stability of the clones. Embryoid formation was detected and characterization of three germ layers (endoderm, ectoderm, and mesoderm) via detection of gene expression of ectoderm (GATA4 & GATA6), mesoderm (RUNX1 & ACTA2), and endoderm (TUBB3 & PAX6) was performed. These markers were assessed by RT-PCR (Figure 2h).

### 3.3. Recellularized Rat Liver Scaffold Characterization

To further characterize liver recellularization and hepatocyte differentiation, gene expression profiles were assessed in three stages: endodermal induction, hepatic specification, and hepatocytes maturation.

Endodermal induction was characterized by significant downregulation of the pluripotent genes (Nanog, SOX2 & OCT4) in bio-scaffold culture (Figure 3a) accompanied by significant up-regulation of FOXA1, FOXA2, SOX17 after 4 days (group IIIa). This was followed by significant downregulation at 14 days (group IIIb, hepatic specification) and 24 days (group IIIc, hepatocytes maturation) in comparison to the native liver (group I; control group) (Figure 3b).

Expression of Wnt pathway related genes changed among various groups, reflecting variability of wnt signaling between various stages of liver maturation. Indeed, Wnt ligand genes (Wnt1, Wnt2, Wnt3a, Wnt7b, Wnt10b) exhibited significantly decreased expression in group IIIa when compared to the native liver (control group) (Figure 3c). Wnt1 and Wnt2 expression levels rose significantly in both groups IIIb and IIIc). In contrast to other Wnt family members, Wnt3a, Wnt7b, and Wnt10b revealed significant upregulation in group IIIb while group IIIc did not. HEX and HLX displayed a similar transcript expression cycle to Wnt3a, 7b, and 10b being upregulated in group IIIb but not thereafter, whereas Prox1, ALB, and CK19 displayed significant increases in both group IIIb, and to a lesser extent in group IIIc (Figure 3d). Expression levels of AFP were significantly reduced in groups GIIIA and GIIIC when compared to control.

To evaluate extent of growth factors expression in various stages of liver maturation, expression of HGF, FGF4, EGF, FGF2, BMP4, and TGF-β was assessed. As displayed in Figure 3e, the aforementioned growth factors showed significant upregulation across groups IIIa, IIIb, and IIIc, and to the greatest extent in group IIIb.

Finally, we explored the expression of markers associated with hepatoblast maturation across the group III subsets. Interestingly, C/EBPα, HNF-4α, PECAM1, and EpCAM displayed gradual increase across the group III subsets with the highest and significant upregulation in group IIIc (Figure 3f). Coupled with the above, we also found significant upregulation of liver-specific microRNAs like miR122, miR148a, and miR194 in group IIIc. Taken together, we observed a broad temporal upregulated expression profile of transcripts associated with induction (group IIIa), specification (group IIIb), and maturation (group IIIb) (Figure 3g), characterizing liver recellularization and hepatocyte differentiation.

### 3.4. Albumin and Urea Secretion by Recellularised Livers

Albumin and urea were below the threshold for detection in perfused culture media after 4 days (group IIIa). After 14 days (group IIIb) albumin and urea synthesis had reached detectable levels of 120 µg/mg and 260 µg/mg, respectively. In group IIIc, albumin and urea levels increased further to group IIIb reaching 200 µg/mg and 400 µg/mg, respectively (Figure 3h) and supporting the evidence suggestive of hepatocyte maturation.

### 3.5. Protein Expression of Albumin, AFP and β-Catenin

Western blot anlysis for Albumin and AFP showed significant decrease in GIIIA (4 days recellularization) compared to GI (native liver). However, GIIIB and GIIIC (14, 24 days recellularization) showed marked restoration of hepatic Albumin and progressive decreae in AFP protein levels. Comparison of total β-catenin in the native liver (group I) with the three stages of differentiation revealed a significant elevation in group IIIB (14 days recellularization) and decreased expression levels in other groups. In contrast to total protein, the phosphorylated β-catenin was down-regulated in all recellularized groups vs. native liver group GI. However, activated β-catenin, similar to total protein, displayed significant elevation in group IIIB (14 days recellularization) only, aligning strongly with ongoing hepatoblasts specification (Figure 4).

### 3.6. Histological Analysis of Recellularized Liver Bio-Scaffolds

#### 3.6.1. Light Microscopy

##### Hematoxylin and Eosin Staining

Group I displayed cords of hepatocytes radiating from a central vein, each a single cell thick, which bifurcated and fused into a network. Hepatocytes were broadly cubical with an acidophilic cytoplasm, central basophilic nuclei, and blood sinusoids present between the cords of hepatocytes (Figure 5a). Group II (decellularized) revealed no visible cell nuclei or material (Figure 5b). In group III (recellularized groups) there was a minimal distribution of cellular engraftment across the scaffold at day 4 (group IIIa) with moderate repopulation of cells after 14 days (group IIIb), and a complete repopulation at day 24 (group IIIc) which saw cells repopulating the empty hepatic spaces and forming colonies resembling hepatocytes as maturing hepatocytes organize into chord-like structures (Figure 5c–e).

##### Masson’s Trichrome

Group I (control group) displayed an absence of collagen fibers around the central vein with small amounts at the portal tract (Figure 6a). Group II displayed preservation of honeycomb appearance collagen fibers and the portal and lobular connective tissue structures (Figure 6b). Recellularized groups (IIIb, IIIc, IIId) displayed gradual coverage of collagen fibers by cell colonies to form complete sheets of hepatocyte-like cells at day 24 (Figure 6c–e). Quantification of the area percentage of collagen fibers indicated a significant increase in group II, vs. control group reflecting the removal of the parenchymal cells. The recellularized group (group IIIa) showed no significant difference compared to group II but remained significantly greater than the control, indicating that the introduced cells remained insufficient to cover the collagen fibers. A similar pattern was observed for group IIIb while group IIIc was comparable to control indicating increased coverage by the seeded cells (Figure 6f).

#### 3.6.2. Ultrastructure Highlights the Capability of Perfused iPSCs to Differentiate and Recellularize the 3D Acellular Bio-Scaffolds with Hepatocyte Property Cells

Normal liver revealed segmented liver parenchyma with regular interdigitating portal tracts and hepatic veins with a branching structure. Hepatocellular plates appeared polyhedral with multifaceted hepatocytes forming distinct one-cell thick plates (laminae). Plates were continuous excepting holes for sinusoidal penetration (lacunae). Neighboring plates were separated by sinusoids (Figure 7a,b). Decellularized livers, in confirmation of above, displayed a complete absence of cells with remaining scaffolds consisting of a three-dimensional network of connective tissue fibers arranged in a honeycomb-like structure (Figure 7c). In recellularized groups, there was a gradual increase in cell density and reduction of hepatocyte pockets ultimately forming cells sheets that covered all collagen bundles (Figure 7d–f). Quantification of pockets showed significant increases in groups IIIa and IIIb when compared to the control group (native liver) indicating decreased cell density. Group IIIc, consistent with Masson’s Trichrome results, showed no significant difference when compared to the control group (Figure 7g).

## 4. Discussion

Bioengineered liver grafts could provide an alternative treatment strategy for patients suffering from terminal stages of liver diseases. Tissue-engineered liver utilizing decellularized ECM as a scaffold reflects a recent strategy in regenerative medicine approaches for organ replacement. Principally the ECM composition must include the physical and chemical properties that meet the requirements of the ultimate clinical usage [28,29]. Decellularization seeks to eliminate cellular components while preserving the biochemical, structural, and mechanical properties of the ECM scaffold. Consequently, liver regeneration based on perfused decellularized native liver ECM scaffolds could provide promise for patients suffering from terminal liver disorders. However, several challenges must be addressed to permit its clinical translation [30].

Previous reports utilized decellularization and subsequent reseeding, via perfusion, with alternate cell types (MSCs, hepatocytes progenitors, and pluripotent stem cells) to manufacture artificial liver grafts [31,32]. Furthermore, several entire organ decellularization protocols have been described, where each had varying effects on the ECM [33,34]. Herein, we utilized a slightly disruptive procedure for entire liver decellularization in a rat model that generated a 3D-liver matrix appropriate for supporting functional hepatocytes. This was achieved via modification of a perfusion-based decellularized heart protocol [35] and a previous report rat liver decellularization [36]. We incorporated freeze-thaw processing to enable effective cell lysis with minimal disruption to the ECM ultrastructure [37].

The Triton/SDS approach provided preservation of the main ECM proteins including laminin, collagen I, collagen IV, and fibronectin. Furthermore, this protocol preserved vascular tree structures minimizing oxygen diffusion distance to the cells and providing an effective technique for the delivery of decellularizing agents to cells and the conveyance of cellular material from the tissue. In agreement, following decellularization protocol in the current study, the liver became transparent, and the three-dimensional microarchitecture was preserved. This was confirmed by histological examination which revealed that collagen fibers were preserved in the decellularized scaffold with small lobular components demonstrated by Masson’s Trichrome staining. Likewise, SEM showed the overall liver tissue was maintained with the microvascular network of the liver present post decellularization. Moreover, Western blot analysis revealed preservation of collagen type IV, fibronectin, and laminin besides preservation of GAG concentration, indicating a similar structural composition of the matrix to that of intact rat liver. This proposed that the matrix of both the sinusoids (collagen type IV, fibronectin) and large vessels (laminin) remained intact. Furthermore, DNA, commonly used as a key marker of decellularization, was significantly decreased below that of the normal liver to the levels lower than those indicated in earlier studies [13,34]. Moreover, in the present study, the decellularized liver matrix preserved about 13–20% of growth factors including HGF, bFGF, VEGF, and IGF-1 when compared to the normal liver. These preserved growth factors potentially provide environmental signals in the ECM to perfused cells for efficient survival and function [38]. Additionally, these factors are crucial for hepatic differentiation, regeneration, and even neoangiogenesis [34].

Stem cells derived from their own patient cells could be the perfect source to engineer personalized liver grafts. Thus, in the current study, the decellularized rat liver scaffolds reseeded with iPSCs cells to form the metabolically functional entire liver construct. Numerous procedures utilized growth factors throughout the hepatogenesis process to improve stem cells’ differentiation capacity into hepatocyte-like cells [39,40]. Moreover, the definitive endoderm specification into hepatocytes would be competently prompted by adding high concentrations of Activin A accompanied by Wnt3a, FGF2, HGF, or BMP4 [41,42,43]. Thus, in the current study, we co-cultured the iPSCs in the presence of growth factors necessary for endodermal and hepatoblast specification, hepatocyte differentiation, and finally hepatocyte maturation. This multi-stepped approach used a biochemically defined system based on the signaling pathways involved in the process of hepatocyte differentiation and maturation to generate functional hepatocytes.

The cultured iPSCs in the 3D bio-scaffold exert high cell-cell interactions through the tight gap junctions, which are readily attached to the ECM via the combination between matrix and cell adhesion molecules, mainly N-cadherin and N-CAM. Moreover, as the differentiation process of iPSCs progressed, the expression of focal adhesion receptors to early pericellular matrix molecules, like fibronectin, was reduced while PECAM1 expression was increased [38]. Similarly, in the current study, there was significant upregulation of EpCAM and PECAM1 with iPSCs morphological changes progressed from a flattened fibroblastic-like shape to the cuboidal morphology of hepatocytes. These morphological changes were synchronized with the reorganization of the cytoskeleton and the downregulation of the stem cell markers like OCT-4, NANOG, and SOX2 that have key roles in the maintenance of pluripotency and self-regulation.

The significant downregulation of the pluripotent stem cell markers was associated with significantly increased expression of FOXA1, FOXA2, and SOX17, indicating endodermal specification of iPSC towards the expression of hepatic genes with subsequent significant upregulation of wnt ligands genes (Wnt1, Wnt2, Wnt3a, Wnt7b and Wnt10b) and increased activated b-catenin and decreased phosphorylated b-catenin, indicating activation of the wnt/b-catenin pathway in recellularized liver. These results were concomitant with Apte et al. [19] who stated that b-catenin is a key constituent of the Wnt pathway which plays a significant role in growth regulation throughout liver development, regeneration, and in ex-vivo embryonic liver cultures.

Moreover, in the current study, there was significant upregulation of FGF, BMP4, and HGF at this stage in the recellularized liver. Concomitant with these results, Jung et al. [44] revealed that FGF is essential for the morphogenetic outgrowth of the liver. Additionally, Rossi et al. [45] revealed that BMP4 signaling is essential for hepatogenesis. Besides, HGF is critical for this stage of liver growth as well as all stages of hepatogenesis.

The differentiated iPSCs within the bio-scaffolds were considered at this stage as hepatoblasts exerting bipotential capability, which meant that they were able of giving rise to both chief lineages of the liver hepatocytes and cholangiocytes. As in the current study, the three multi-stepped culture of iPSCs within the 3D bio-scaffold revealed a significant reduction in gene expression of fetal liver genes (AFP) and significantly increased levels of albumin, which is a marker of hepatocytes with significant increased cytokeratin 19 (CK19) which is a cholangiocytes marker. These results confirmed the bipotentiality of hepatoblasts which could differentiate into hepatocytes or cholangiocytes. Furthermore, there was a significantly increased gene expression of HGF and BMP4 in all recellularized groups. Additional transcription factors needed for the hepatocytes specification at this phase include HLX58 and Prox1. Consistently, the results of the current study revealed significant upregulation of Hex, HLX, and Prox genes expression in all recellularized groups. The aforementioned genes are crucial for liver proliferation and differentiation. Interestingly, these genes have been reported as downstream targets of the β-catenin pathway [15]. Notably, the canonical Wnt pathway is activated by Wnt ligand binding leading to stabilization of cytosolic β-catenin. Stabilized β-catenin then translocates into the nucleus, where it activates target genes crucial for proliferation and differentiation. Downstream targets of β-catenin, such as c-Myc and cyclin, prompt proliferation, while others, such as BMP4, Hex, HNF-4, and C/EBPα, stimulate differentiation [43,46].

These results are consistent with Nejak-Bowen K. and Monga S. P. S. (2008) [47] who noted that repression of β-catenin (increased phosphorylated β-catenin) in the endodermal stage is essential for liver development, and a trigger for formation of liver markers. On the other hand, β-catenin expression is required during liver specification and expression of Hex and Prox1, both essential in hepatoblast formation, implicating these transcription factors as potential downstream targets of the β-catenin pathway [21]. Therefore, Wnt/ β-catenin pathway is suggested to be a positive regulator of liver specification. Indeed, hepatoblast specification is accompanied by increased β-catenin protein, during which time it is localized throughout cell compartments including the nucleus, cytoplasm, and membrane. Hepatocytes maturation is accompanied by decreases in activated β-catenin gene expression and decreased total β-catenin protein expression. Positive correlation between β-catenin and cell proliferation have been demonstrated in various studies [47] and is linked to cell cycle mediators such as cyclin D1, itself a known downstream target of β-catenin. The presence of β-catenin, as well as its location inside the cell, may therefore play a key role in dictating cellular differentiation [48].

Mechanistic studies of Wnt/β-catenin activation in liver development have implicated FGF proteins as upstream mediators of this pathway. Positive corrections have been reported between FGF-10 expression and β-catenin activation. Moreover, β-catenin expression in hepatoblasts is stimulated by release of FGF-10 from stellate cells [49]. Furthermore, FGF-2, FGF-4 and FGF-8 have been reported to promote an increase in the number of hepatic progenitor cells in ex vivo embryonic livers. Interestingly, hepatic β-catenin expression is induced by addition of these growth factors to explanted livers, suggesting FGFs as activators of the Wnt pathway, where FGF-8 is a known downstream target of the Wnt pathway [21].

Therefore, plenty of evidence demonstrated crucial role of Wnt/β-catenin signaling in liver development [50]. Livers from mouse embryos cultured in the presence of a β-catenin antisense oligonucleotide demonstrated a reduction in proliferation with a simultaneous increment of apoptosis [51]. Additionally, blockage of β-catenin expression through overexpression of pathway inhibitors led to decreased liver size with alteration of liver shape [49]. This data supports a highly temporal expression and activation of the Wnt/β-catenin signaling pathway during the process of normal liver development and any deviation from this norm could lead to abnormal growth with its consequences.

The last stage of recellularization was characterized by differentiation of hepatoblasts into mature, fully functional hepatocytes as mature hepatocytes start to organize into cord-like structures in part through stimulation by HGF. At this differentiation process, the liver bio-scaffold was enriched with transcription factors such as HNF, which controls cell fate decisions. HFN-4α is considered vital for differentiation into a hepatocyte phenotype, in addition to the development of the parenchyma [52]. This was in line with the results of the current study as there was a significant downregulation in the expression of AFP with significant upregulation in the expression of HFN-4α and C/EBPα demonstrating hepatocyte maturation. This was concomitant with progressive increase in functional ability as evidenced by enhanced albumin secretion as well as urea synthesis in culture media. Moreover, β-catenin’s role in cell-cell adhesion may also be vital for hepatocyte maturity, as the association of β-catenin with E-cadherin increases between E16 and E18 in the murine liver [53]. With this context, the current study showed decreased total level of b-catenin with increased phosphorylated than the activated forms of b- catenin in last stage which could be explained by the presence of membrane-associated b-catenin.

## 5. Conclusions

In the present study, perfusion protocol for decellularization generates a three-dimensional organ scaffold that can be used to establish human liver grafts for transplantation using iPSCs recellularizaton. Additionally, current data raised the significance of the Wnt/β-catenin pathway in iPSCs differentiation into hepatocytes (Graphical Abstract).

## Figures and Tables

**Figure 1 cells-10-02819-f001:**
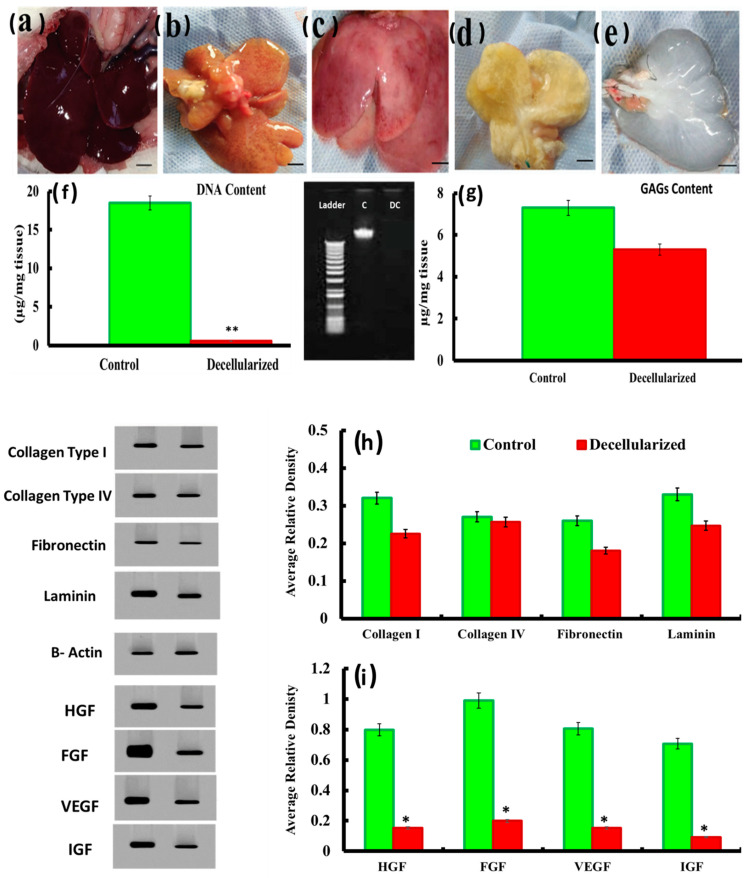
Representative images of livers during decellularization process at (**a**): 0 h; (**b**): 18 h;(**c**): 48 h; (**d**) 72 h, and (**e**): 96 h. (**f**): DNA content analysis; (**g**): glycosaminoglycans (GAGs) analysis of normal liver compared to decellularized (DC) rat liver scaffold; (**h**,**i**): Western blot assay for ECM components & growth factors of normal liver and after decellularization process. β-actin was used as housekeeping protein and quantification was performed using Image analysis software on the Chemi Doc MP imaging system. Superscripts (*) indicate significant differences at *p* ˂ 0.05. Superscript (**) indicates significant differences at *p* ˂ 0.01. Data are shown as mean ± S.E.M, *n =* 7.

**Figure 2 cells-10-02819-f002:**
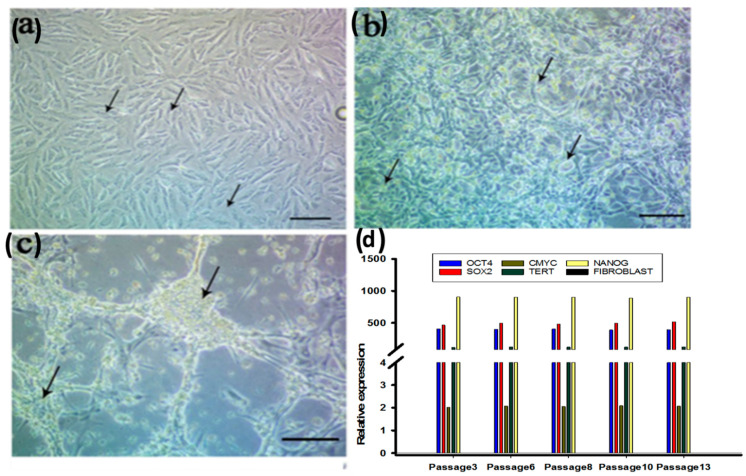
IPSCs characterization showed (**a**,**b**): Dermal fibroblasts cultured at day 0 and expanded for 7 days; (**c**): Representative images of iPSCs colonies at 4 weeks transfection after re-plating and re-expansion; (**d**): RT-PCR assessment for pluripotent genes (OCT4, SOX2,c-MYC, TERT & NANOG) in different 5 IPSCs passages. (**e**): Western blot assay for protein level of OCT4, SOX2 and Nanog. (**f**): Flow cytometry to detect SSEA-4 in iPSCs. (**g**): Embryoid formation with spontaneous tri-linage differentiation; (**h**): qRT-PCR assessment for iPSCs spontaneous differentiation genes for ectoderm (GATA4, GATA6), mesoderm (RUNX1 & ACTA2), and endoderm (TUMM3 & PAX6).

**Figure 3 cells-10-02819-f003:**
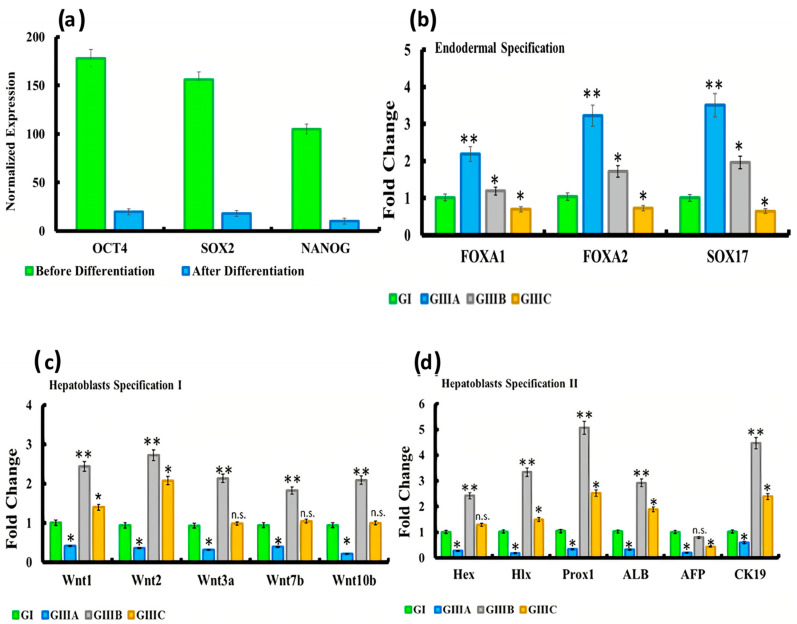
Histogram showed (**a**): Quantitative analysis for relative expression of OCT4, SOX2 & Nanog before the perfusion technique and after iPSCs differentiation within the 3D bio-scaffolds; (**b**–**g**): Quantitative analysis for relative expression of FOXA1, FOXA2, FOX17, HHEX, FGF2, BMP4, HGF, FGF4, EGF, TGF-β, Prox1, CK19, C/EBPα, HNF-4α, AFP, ALB, HLX, PECAM, EpCAM, miR122, miR148a, and miR194 genes after iPS cells implantation. (**h**): Detection of albumin and urea concentration in the perfusion medium by ELISA from different experimental groups. The significant differences against the control group indicated by a * for *p* < 0.05 and a ** for *p* < 0.01. Data are shown as mean ± S.E.M, *n* = 7. GI (native liver), GIIIA (4 days recellularization), GIIIB (14 days recellularization) and GIIIC (24 days recellularization).

**Figure 4 cells-10-02819-f004:**
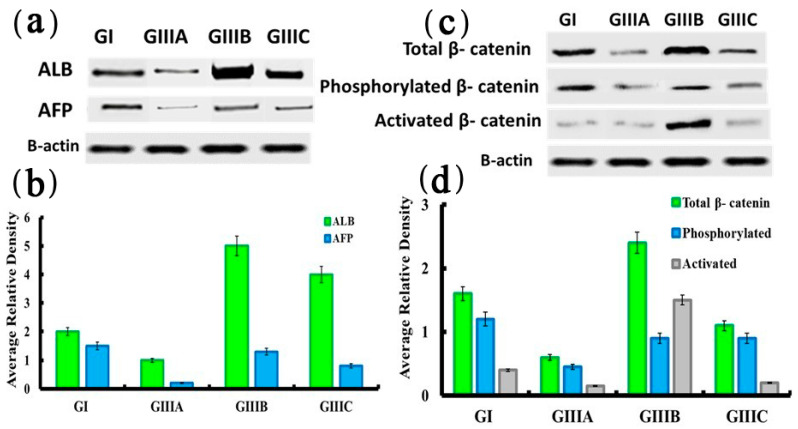
Protein expression of ALB, AFP and β-catenin (total, activated & phosphorylated) by western blot analysis (**a**–**d**). β-actin was used as housekeeping protein and quantification was performed using Image analysis software on the Chemi Doc MP imaging system. Data are shown as mean ± S.E.M, *n =* 7. GI (native liver), GIIIA (4 days recellularization), GIIIB (14 days recellularization) and GIIIC (24 days recellularization).

**Figure 5 cells-10-02819-f005:**
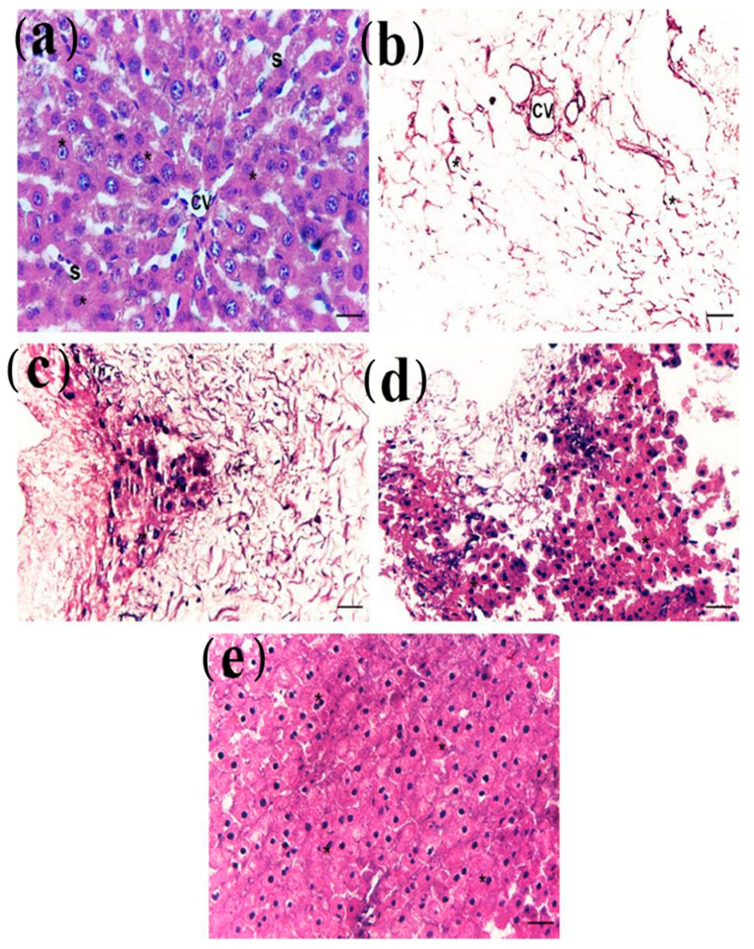
Photomicrograph of a section in normal liver, decellularized scaffolds as well as recellularized scaffold 4, 14, and 24 days after iPSCs cell seeding showed (**a**): group I (control group) showed cords of normal hepatocytes radiating from a central vein (CV), hepatocytes (*) are cubical with a large central basophilic nucleus and blood sinusoids (S); (**b**): group II (decellularized group) showed no visible cell nuclei and cellular material (*) in all decellularized livers; (**c**): group IIIa (recellularized group after 4 days): showed a minimal distribution of cellular engraftment (*) across the scaffold at day 4; (**d**): group IIIb (recellularized group after two weeks): showed a moderate distribution of cellular engraftment (*) across the scaffold at day 14; (**e**): group IIIc (recellularized group after 24 days): showed complete cell sheets (*) forming colonies resembling hepatocytes.

**Figure 6 cells-10-02819-f006:**
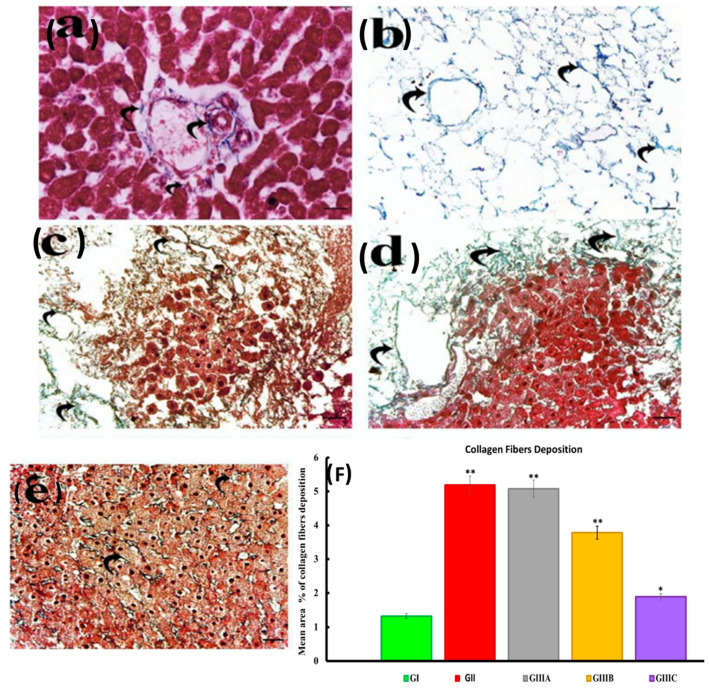
A photomicrograph of Masson’s trichrome stained sections in normal liver, decellularized scaffolds as well as recellularized scaffold 4, 14 and 24 days after iPS cell seeding showed (**a**): group I (control group) showed minimal collagen fibers accumulation (curved arrow) around portal tract; (**b**): group II (decellularized group) showed preserved collagen fibers (curved arrow) as well as the portal and lobular connective tissue structure in decellularized liver scaffolds; (**c**): group IIIa (recellularized group after 4 days) showed marked collagen fibers (curved arrow) with minimal amount of cell colony (*); (**d**): group IIIb (recellularized group after 14 days) showed moderate collagen fibers (curved arrow) with moderate amount of cell colony (*); (**e**): group IIIc (recellularized group after 24 days) showed minimal collagen fibers (curved arrow) with extensive amount of cell colony forming sheets of hepatocytes (*); (**f**): A histogram represents the mean area percentage of collagen in all experimental groups. The significant differences against the control group indicated by a * for *p* < 0.05 and a ** for *p* < 0.01. Data are shown as mean ± S.E.M, *n =* 7.

**Figure 7 cells-10-02819-f007:**
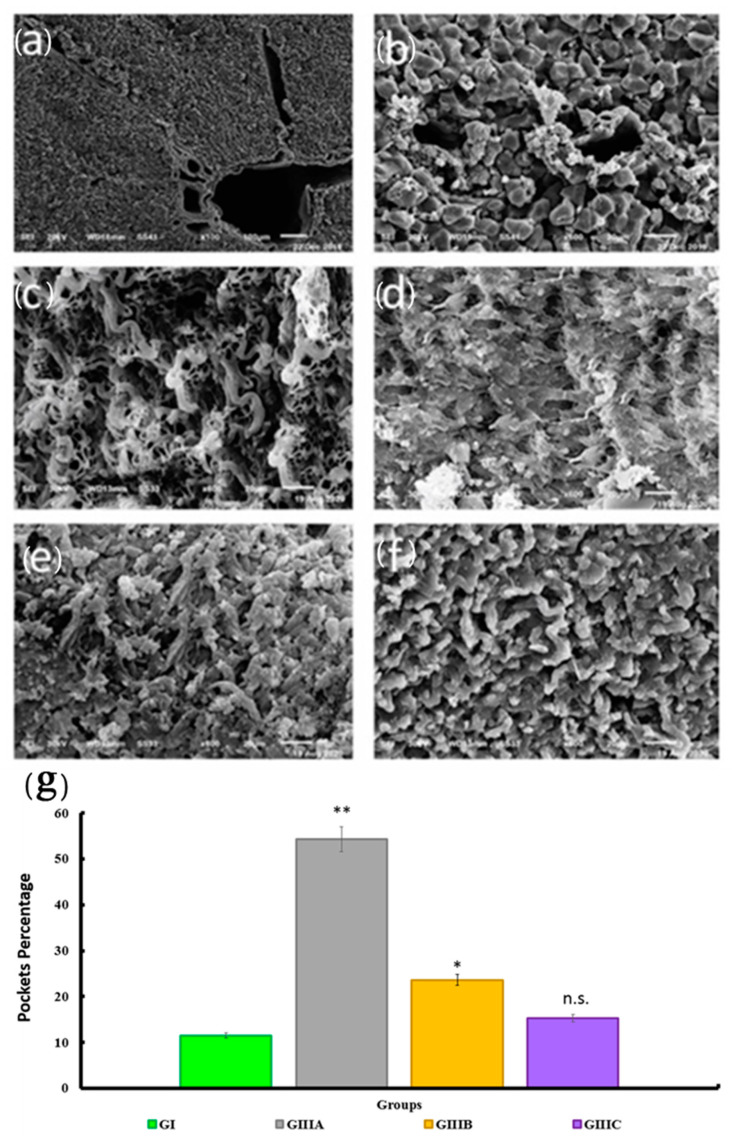
A photomicrograph of control liver, decellularized scaffolds as well as scaffold 4, 14, and 24 days after IPS cell seeding showed (**a**): Microscopic segmentation of hepatic parenchyma, structures such as the hepatic lobules, the central veins (CV), Portal vein (V), hepatic artery (A) and extracellular matrix are present within the parenchyma, x100; (**b**): Ultrastructural characterization of the normal liver matrix showed hepatic plates (*) with the central vein (CV), ×600; (**c**): Ultrastructural characterization of the decellularized scaffold showed rope-like bundles (curved arrow) and filaments of connective tissue fibers forming three-dimensional network arranged in a honeycomb-like structure with hepatocytes pockets (thin arrow), ×600; (**d**–**f**): Ultrastructural characterization of the recellularized scaffold showed decreased hepatocytes pockets (thin arrow) with increased cellular content (*) with the increased period of seeding, ×600. (**g**): A histogram represents the mean pocket number in all experimental groups. The significant differences against the control group indicated by a * for *p* < 0.05 and a ** for *p* < 0.01. n.s.: non-significant. Data are shown as mean ± S.E.M, *n =* 7.

**Table 1 cells-10-02819-t001:** Sequence of primers used in RT-PCR.

Gene	Accession №	Primers Sequences (5′→3′)		Amplicons
FOXA1	NM_012742.1	F:	GACGTTCAAGCGCAGTTACC	189
		R:	GACAGTGAGTGGCGAATGGA	
FOXA2	NM_012743.1	F:	CTGAGGTGGGTAGCCAGAAAAA	160
		R:	CACGGCTCCCAGCATACTTT	
SOX17	NM_001107902.1	F:	TTCAGCCGTCCTATTTCCCC	187
		R:	CTGGTCGTCACTGGCGTATC	
FGF2	NM_019305.2	F:	TCCATCAAGGGAGTGTGTGC	139
		R:	TCCGTGACCGGTAAGTGTTG	
BMP4	NM_012827.2	F:	CAGGGCCAACATGTCAGGAT	188
		R:	TGGCGACGGCAGTTCTTATT	
HGF	NM_017017.2	F:	ACAGCTTTTTGCCTTCGAGC	178
		R:	TAGCTTTCACCGTTGCAGGT	
FGF4	NM_053809.1	F:	CTACCTGCTGGGCCTCAAAA	130
		R:	CACACCCCGCTGCTGTC	
EGF	NM_012842.1	F:	CGATGTCAGCACCGAGACTT	202
		R:	CGTTGCTGCTTGACTCTTCG	
TGF-β	NM_021578.2	F:	GACTCTCCACCTGCAAGACC	100
		R:	GGACTGGCGAGCCTTAGTTT	
ALB	NM_134326.2	F:	TCGTATGAGCCAGCGATTCC	159
		R:	GTGGCCTGGTTCTCACACAT	
AFP	NM_012493.2	F:	CCAGTGCCCGACAGAGAAAA	120
		R:	TTCATTGCAGCCAACGCATC	
CK19	NM_199498.2	F:	TTGGGTCAGGGGGTGTTTTC	208
		R:	AGGCGATCGTTCAGGTTCTG	
HHEX	NM_024385.1	F:	CAGCGACCTCTGCACAAAAG	128
		R:	ATCTTGGCCAGACGCTTTCT	
HLX	NM_001077674.1	F:	CTGGCTCCCTTCTACGCTTC	131
		R:	ATGTCCGCGATGCAGAAAGA	
Prox1	NM_001107201.1	F:	TTGACTCGGGACACAACGAG	191
		R:	TGATAGCCCTTCATTGCGCT	
C/EBPα	NM_001287577.1	F:	GGGAGCAAACATGTGCCTTG	168
		R:	TCTAAGGACAGGGACGGAGG	
HNF-4α	NM_001270931.1	F:	ATGAGCTGGTCTTGCCCTTC	183
		R:	GAGAGTCATACTGCCGGTCG	
PECAM1 (CD31)	NM_031591.1	F:	GGTAATAGCCCCGGTGGATG	160
		R:	TTCTTCGTGGAAGGGTCTGC	
EpCAM	NM_138541.1	F:	CGATCCAGAACAACGACGGT	135
		R:	CCGTGTCCTTGTCGGTTCTC	
*SOX2*	NM_001109181.1	F:	CTCTGTGGTCAAGTCCGAGG	105
		R:	ATGCTGATCATGTCCCGGAG	
OCT4	XM_032889059.1	F:	CCTGGGCGTTCTCTTTGGAAAGGTG	198
		R:	GCCTGCACCAGGGTCTCCGA	
Nanog	NM_001100781.1	F:	ACACACCCACCCTACTCCAT	174
		R:	ACGATACACAGTGCACACCA	
Wnt1	NM_001105714.1	F	CGTTGCTGTCCCTGTGGTAT	105
		R	CAGGTGTGGTGGTTAGGGAC	
Wnt2	XM_575397.8	F	CAGAATGCGTGGGCTAGTCA	91
		R	TCACCCTTGGAATGGATGGC	
Wnt3a	NM_001107005.2	F	CGGGTTCTTCTCTGGTCCTTG	218
		R	CTGACAGTGGTGCAGTTCCA	
Wnt7b	NM_001009695.1	F	CTCTGCTTTGGCGTCCTCTAC	184
		R	GCTGGCATTCATCGATACCC	
Wnt10b	NM_001108111.1	F	CCCTGTCCGGCTTGAGTAAG	214
		R	AAGGAGAACGCACTCTCACG	
TUBB3	NM_139254.2	F	CAACTATGTGGGGGACTCGG	89
		R	TGGCTCTGGGCACATACTTG	
FOXA2	NM_012743.2	F	AGGTGGGTAGCCAGAAAAAGG	197
		R	AGTAGCTGCTCCAGTCGGAT	
RUNX1	NM_017325.2	F	CCAAACTCTGAAAGCGAGGC	88
		R	TTAGCAACTGGCCGCTTAGT	
ACTA2	NM_031004.2	F	ACCATCGGGAATGAACGCTT	191
		R	CTGTCAGCAATGCCTGGGTA	
GATA6	NM_019185.2	F	GCCAACCCTGAGAACAGTGA	166
		R	GTATGAGGCCTTCAGAGCCC	
GATA4	XM_017599788.2	F	ACCCATCACACAGATCGCAG	85
		R	TGTTCAGGCTGGAGAGCAAG	
PAX6	NM_013001.2	F	CCGAATTCTGCAGGTGTCCA	111
		R	GTCGCCACTCTTGGCTTACT	
TERT	NM_053423.1	F	TTCCTTCCACCAGGTGTCATC	88
		R	AGCCAGCACATTCCTCTCAC	
c-Myc	NM_012603.2	F	TGAAAAGAGCTCCTCGCGTT	139
		R	AAATAGGGCTGCACCGAGTC	
B-Actin	NM_031144.3	F:	GCGAGTACAACCTTCTTGCAG	70
		R:	TCGTCATCCATGGCGAACTG	

## Data Availability

All data generated and/or analyzed during this study are included in this published article.

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
