# Peer review of "Functional Recellularization of Acellular Rat Liver Scaffold by Induced Pluripotent Stem Cells: Molecular Evidence for Wnt/B-Catenin Upregulation"

_cells, 2021, doi:10.3390/cells10112819_

Round 1

Reviewer 1 Report

The manuscript “Role of Wnt/β catenin Pathway in Functional Recellularization of Acellular Rat Liver Scaffold by Induced Pluripotent Stem Cell” is focusing on the development of a 3D organ scaffold using hepatocytes differentiated from iPSCs. The potential use of the scaffold generated is interesting in the field of liver diseases and transplantation. Authors used as a scaffold a decellularized liver thus representing biocompatible support for the transplantation of hepatocytes.  However, it has been already demonstrated that it is possible to generate organoids or to transplant hepatocytes in a bio-scaffold and the Wnt/β catenin Pathway is not the focus of the paper even if listed in the title of the manuscript.

Revision:

However, major revisions are required:

  • iPSCs were characterized only at P1 and P2. Usually, iPSCs clones should be stable and characterization at higher passages is required.
  • iPSCs were characterized for 3 pluripotency markers at RNA level. But a deeper characterization is required (more markers and protein level).
  • EBs were characterized using just one marker for each germ layer. Some more markers are needed.
  • All the characterization process of described cells has been very superficial 
  • All the markers of hepatocytes are at RNA level, but some of the most important (ALB, AFP) need to be analyzed also at the protein level.
  • Authors explain that AFP and ALB levels are significantly increased in some conditions. However, AFP has a very low increase in fold change (around 0.5). It should be reported the reference sample on which has been calculated the fold change.
  • In the western blot ptresent in fig.4 the legend is missing, thus it is not clear what are the samples. Moreover, the figure should be edited or corrected it seems a puzzle of several pictures,it is better to increase the overall quality. Finally, it is not clear why the levels of total beta-catenin are changing in the different conditions and this has not been explained anywhere. This affects the understanding of the phosphorylation and the levels of active beta catenin.

Author Response

  • iPSCs were characterized only at P1 and P2. Usually, iPSCs clones should be stable and characterization at higher passages is required.

Thank you for your comment. In our lab, we have already characterized the iPSCs in different passages to examine its maturation and stability. So, we performed PCR analysis to confirm the induction of endogenous pluripotency-associated markers including OCT4, SOX2, NANOG, TERT, and c-MYC. The markers of pluripotency remained unchanged from passages 3 to 13 (Figure 2d), suggesting the maintenance of an undifferentiated state of the iPSCs clones produced, thus confirming the stability of the clones.

  • iPSCs were characterized for 3 pluripotency markers at RNA level. But a deeper characterization is required (more markers and protein level).

Thank you for your suggestion. The iPSCs were not only confirmed morphologically, but we also assessed the transcriptional level for a number of commonly observed pluripotency markers OCT4, Nanog, Sox2, c-MYC and TERT demonstrating that the cells were maintained in a state of pluripotency. Furthermore, the western blot assay for OCT-4, Nanog &SOX-2 to assess their protein level in iPSCs line were done (Figure 2e.). Finally, we performed flow cytometry analysis for some of the markers (SSEA-4) confirming the pluripotency of the clones (Figure 2f).

  • EBs were characterized using just one marker for each germ layer. Some more markers are needed.

Thank you for your comment. Under the appropriate conditions, the iPSCs readily formed embryoid bodies and differentiated in vitro into cells expressing markers of ectoderm (GATA4& GATA6), mesoderm (RUNX1& ACTA2), and endoderm (TUBB3& PAX6). These markers were assessed by RT-PCR (Figure 2h) 

  • All the characterization process of described cells has been very superficial.

We increased some markers for iPSCs characterization OCT4, Nanog, Sox2, c-MYC and TERT.

Plus, for three linage differentiation we assessed for ectoderm (GATA4, GATA6), mesoderm (RUNX1& ACTA2), and endoderm (TUMM3& PAX6).

  • All the markers of hepatocytes are at RNA level, but some of the most important (ALB, AFP) need to be analyzed also at the protein level.

Thank you for your comment. The assessment of ALB, AFP at protein level was performed (Figure 4).

-Authors explain that AFP and ALB levels are significantly increased in some conditions. However, AFP has a very low increase in fold change (around 0.5). It should be reported the reference sample on which has been calculated the fold change.

We used the sample with the highest Ct value i.e. the sample with the lowest gene expression ΔΔct= Δct of treated sample - ΔCt of control.

ΔCt=28.47-18.38=10.09

ΔCt=26.68-17.88=8.8

ΔΔct=10.09-8.8=1.29

So, on applying the formula 2^-(ΔΔct)=2^-(1.29)=0.408

  • In the western blot present in fig.4 the legend is missing, thus it is not clear what are the samples. Moreover, the figure should be edited or corrected it seems a puzzle of several pictures, it is better to increase the overall quality. Finally, it is not clear why the levels of total beta-catenin are changing in the different conditions and this has not been explained anywhere. This affects the understanding of the phosphorylation and the levels of active beta catenin.

Thank you for your comment. Figure legend has been added to figure 4 and a new figure with improved quality was added.

To explain the part of b-catenin we added two part in the text (1st part in the introduction) and the 2nd part in the discussion.

Introduction:

AS the Canonical Wnt signaling was inactive in the absence of Wnt ligands (Wnt off). In this state, β-catenin is located in adherent junctions and cytoplasm of the cell, where it becomes phosphorylated by the destruction complex (comprising adenomatous polyposis coli protein (APC), Axin, casein kinase I isoform-α (CK1α) and glycogen synthase kinase 3β (GSK3β) and targeted for proteasomal degradation.   While, the Wnt signaling was active in the presence of Wnt ligands (Wnt on), which bind to the FZD–LRP5–LRP6 co-receptor. Subsequent LRP6 phosphorylation then leads to Axin and Dishevelled (DVL) recruitment, which blocks Axin-mediated phosphorylation of β-catenin and thereby prevents β-catenin degradation, enabling its accumulation and nuclear translocation. In the nucleus, β-catenin binds to diverse co-effectors, regulating the expression of genes involved in different cellular processes [20].

Discussion:

These results are concomitant with Nejak-Bowen and Monga [47] who stated that regarding changing the levels of beta-catenin in the different conditions, repressing β-catenin (increased phosphorylated β-catenin) in the endodermal stage is essential for liver development and triggers formation of liver markers. On the other hand, β-catenin expression is required during the liver specification stage and expression of Hex and Prox1, which are essential in hepatoblast formation, implicating these transcription factors as potential downstream targets of the β-catenin pathway [47]. Therefore, Wnt/ β-catenin pathway is suggested as a positive regulator of liver specification. Indeed, hepatoblast specification is accompanied by increased protein expression of β-catenin, during which time it is localized throughout the cell including the nucleus, cytoplasm and membrane. Subsequent decreases in activated β-catenin gene expression and dramatic decrease in total β-catenin protein expression occurs during hepatocytes maturation. Also a positive correlation between β-catenin and cell proliferation have been demonstrated in various studies [48] and has been linked to cell cycle mediators such as cyclin D1, which is a known downstream target of β-catenin. All these studies imply that the presence of β-catenin, as well as its location inside the cell, might be a critical event dictating cellular differentiation [49].

Additionally, mechanistic studies of Wnt/β-catenin activation in liver development have implicated FGF proteins as upstream mediators of this pathway. Indeed, positive correction has been reported between FGF-10 expression and β-catenin activation. Moreover, β-catenin expression in hepatoblasts is stimulated by release of FGF-10 from stellate cells [50]. Further, FGF-2, FGF-4 and FGF-8 have been reported to promote an increase in the number of hepatic progenitor cells in ex vivo embryonic livers. Interestingly, hepatic β-catenin expression is induced by addition of these growth factors to explanted livers, suggesting FGFs as activators of the Wnt pathway. Since FGF-8 is also a downstream target of the Wnt pathway [47].

  1. Kari Nejak-Bowen1 and Satdarshan P.S. Monga1. Wnt/beta-catenin signaling in hepatic organogenesis. [Organogenesis 4:2, 92-99; April/May/June 2008]; ©2008 Landes Bioscience.
  2. Decaens T, Godard C, de Reynies A, Rickman DS, Tronche F, Couty JP, Perret C, Colnot S. Stabilization of beta-catenin affects mouse embryonic liver growth and hepatoblast fate. Hepatology 2007.
  3. Burke ZD, Tosh D. The Wnt/beta-catenin pathway: master regulator of liver zonation? Bioessays 2006; 28:1072-7.
  4. Berg T, Rountree CB, Lee L, Estrada J, Sala FG, Choe A, Veltmaat JM, De Langhe S, Lee R, Tsukamoto H, Crooks GM, Bellusci S, Wang KS. Fibroblast growth factor 10 is critical for liver growth during embryogenesis and controls hepatoblast survival via beta-catenin activation. Hepatology 2007; 46:1187-97.

Reviewer 2 Report

In this original research manuscript, Ebrahim et al. describe the decellularization of rat livers, their subsequent recellularization with rat iPSCs, and the functional differentiation of the pluripotent cells into hepatocytes as evidenced by formation of typical structures and secretion of albumin/urea. The study also aimed to examine the activation of the Wnt/beta-catenin pathway in the stepwise differentiation process.

While the manuscript is of sufficient quality, there are some deficiencies that would need to be rectified in order to be considered for publication:

  1. Particularly, the material and method section concerning iPSC generation is not complete and does not contain all the necessary information.
  • It is not clear what plasmid was used for the reprogramming and where it was acquired from. The only thing that can be inferred from the text that it is a non-integrating plasmid (on line 210, the plural “plasmids” was used, but later it was stated on line 214 that it was a single plasmid). If the authors generated the plasmid themselves, they need to describe the process in detail. Otherwise, please state where the plasmid was obtained from and give credit to the people who generated it.
  • The authors seem to erroneously use the term “reprogramming” instead of “transfection” on lines 208, 215, and 218 – please correct the text.
  1. The characterization of the generated rat iPSCs also raises some questions:
  • The cells were cultured for only 2 passages. Usually, the reprogramming transgenes are still retained in the cells at this time and may still be detected by RT-PCR or immunostaining. This is why it is important to verify, in order to prove that the cells have been fully reprogrammed, whether the endogenous pluripotency genes have been upregulated. In the current case, it is not clear whether the OCT4, SOX2, and NANOG are expressed from the transgenes or from the cell’s own DNA. Use of endogenous gene-specific primers and performing a parallel negative control reaction showing that these primers do not amplify the reprogramming transgenes is necessary.
  • The nucleotide sequence of NANOG cannot be found in Table 1. Please, add the missing information.
  1. One serious problem in the evaluation of the manuscript is the blocking of the figure legends by the figures 1,2,4, and 5. The legends of these figures are not readable.
  2. The fonts used in the graphics is too small and need to be increased in size.
  3. The authors show that the ratios of phosphorylated vs activated beta-catenin changed during the differentiation/maturation process (Fig. 4). However, it is not clear from the material and method section how the phosphorylated and activated beta-catenin were distinguished. There is only one antibody listed, which would detect the total beta-catenin, but no antibody against the phosphorylated version is mentioned. Please, complete the necessary information.

Overall, the manuscript requires a minor revision before it can be accepted for publication.

Author Response

  1. Particularly, the material and method section concerning iPSC generation is not complete and does not contain all the necessary information. It is not clear what plasmid was used for the reprogramming and where it was acquired from. The only thing that can be inferred from the text that it is a non-integrating plasmid (on line 210, the plural “plasmids” was used, but later it was stated on line 214 that it was a single plasmid). If the authors generated the plasmid themselves, they need to describe the process in detail. Otherwise, please state where the plasmid was obtained from and give credit to the people who generated it.

Thank you for your comment. We used ProPlasmid TM from GenScript ProBio containing OCT-4, SOX-2, C-MYC &LIN28 and we add it in the text within the methodology section.

  1. The authors seem to erroneously use the term “reprogramming” instead of “transfection” on lines 208, 215, and 218 – please correct the text.

Thank you for this note, the mistake was corrected.

  1. The characterization of the generated rat iPSCs also raises some questions: The cells were cultured for only 2 passages. Usually, the reprogramming transgenes are still retained in the cells at this time and may still be detected by RT-PCR or immunostaining. This is why it is important to verify, in order to prove that the cells have been fully reprogrammed, whether the endogenous pluripotency genes have been upregulated. In the current case, it is not clear whether the OCT4, SOX2, and NANOG are expressed from the transgenes or from the cell’s own DNA. Use of endogenous gene-specific primers and performing a parallel negative control reaction showing that these primers do not amplify the reprogramming transgenes is necessary.

Thank you for this valuable comment. We applied the endogenous gene-specific primers for OCT4, SOX2 as shown in table 1. Moreover, we performed a parallel negative control reaction showing that these primers do not amplify the reprogramming transgenes as shown in figure (2d).

  1. The nucleotide sequence of NANOG cannot be found in Table 1. Please, add the missing information.

Thank you for your comment.  The sequence of Nanog was added to the primers table.

  1. One serious problem in the evaluation of the manuscript is the blocking of the figure legends by the figures 1,2,4, and 5. The legends of these figures are not readable.

Thank you for this valuable comment.  Figure legends were reviewed, rephrased and more details were added.

  1. The fonts used in the graphics is too small and need to be increased in size.

 Thank you for your suggestion. Figures were improved and font size was increased.

  1. The authors show that the ratios of phosphorylated vs activated beta-catenin changed during the differentiation/maturation process (Fig. 4). However, it is not clear from the material and method section how the phosphorylated and activated beta-catenin were distinguished. There is only one antibody listed, which would detect the total beta-catenin, but no antibody against the phosphorylated version is mentioned. Please, complete the necessary information.

Thank you for this comment. We added the missed antibodies to materials and methods section as follow Total β-catenin (ab32572), Activated β-catenin (ab246504), Phosphorelated β-catenin (Y654, CTNNB1, A00004Y654).

Round 2

Reviewer 1 Report

General comment:

Authors of the manuscript “Functional Recellularization of Acellular Rat Liver Scaffold by Induced Pluripotent Stem Cells: Molecular evidence for Wnt/B-Catenin upregulation” answered to the revisions that were proposed.  However, it has been already demonstrated that it is possible to generate organoids or to transplant hepatocytes in a bio-scaffold and the Wnt/β catenin Pathway is not the focus of the paper.

Revision:

In the first revision was asked to:

  • Characterize iPSCs not only at P1 and P2, but to demonstrate that clones should are stable at higher passages: authors added the characterization up to passage 13
  • Characterize iPSCs for more than 3 pluripotency markers at RNA and protein levels: authors added TERT and c-Myc as markers of pluripotency, performed WB for Nanog, Sox2, Oct4 and performed FACS analysis for Ssea-4
  • Characterize EBs using just more than 1 marker for each germ layer: authors added an additional marker for each germ layer
  • Check for hepatic markers also at protein level: authors performed WB for AFP and ALB showing an increase of ALB expression overtime
  • Add the legend for beta catenin WB and edit the image: WB was change and the image quality is increased

Author Response

Dear Editor 

Dear reviewer

Thank you very much for considering our manuscript. We inspected the comments attached in the second round, but all the comments raised by the respected reviewer are reporting our response to the initial comments.

Could you please help us with this issue?

Best regards